# Strongly Interacting Dark Matter admixed Neutron Stars

**Yannick Dengler, Suchita Kulkarni, Axel Maas, Kevin Radl**

*Institute of Physics, NAWI Graz, University of Graz, Universitätsplatz 5, 8010 Graz, Austria*

*E-mail:* yannick.dengler@uni-graz.at, suchita.kulkarni@uni-graz.at, axel.maas@uni-graz.at, kevin.radl@edu.uni-graz.at

ABSTRACT: Dark matter may accumulate in neutron stars given its gravitational interaction and abundance. We investigate the modification of neutron star properties and confront them with the observations in the context of strongly-interacting dark matter scenario, specifically for a QCD-like theory with $G_2$ gauge group for which a first-principles equation-of-state from lattice calculations is available. We study the impact of various observational constraints and modeling of the QCD equation of state on the combined neutron stars. The results indicate that dark matter masses of a few hundred MeV to a few GeV are consistent with the latest observed neutron star properties.

## 1 Introduction

The dark matter hypothesis [1–3] is and remains one of the primary candidates to resolve a multitude of astronomical observations, despite a lack of direct observation. Assuming the existence of dark matter, its gravitational interactions are a given, with only the masses and species composition, if any, as inputs. In addition, indirect evidence supports a sizable dark matter self interaction [4–12], while interactions with standard model matter (in the following called *ordinary*) remain only bounded from above [2, 3]. Concentrating on dark matter gravitational signals is therefore a prudent strategy. However, except for supermassive dark matter particles [13], individual dark matter particles will not create an observable gravitational signal. Thus, only accumulations of dark matter particles will provide relevant signals, requiring an accumulator.

Due to the gravitational interactions, dark matter will necessarily accumulate in neutron stars. Thus, neutron stars make an excellent laboratory for testing such gravitational signals as their properties are determined by gravitational interactions [14–16], and can be tested using direct observations and gravitational wave astronomy. However, it is a-priori unclear whether the quantity of accumulated dark matter is sufficient to produce observable effects. In particular, depending on the age, history, and location of a neutron star, the accumulated dark matter amount can vary wildly from anything upward from

$10^{-3}\%$ of the neutron star mass, but likely will exceed a percent only in the most extreme circumstances [17–43].

Fortunately, we do have a very good overview of global neutron star properties from observations. Among the direct observations, the Neutron star Interior Composition ExploreR (NICER) mission is dedicated to direct observation of neutron star properties via observations in the soft (0.2–12 keV) X-ray band [44]. The mission concentrates on measurements of neutron star mass and radius, and will provide in its runtime a wide range of characterized neutron stars. Thus, a stochastically large enough sample is available to also detect variations of neutron star properties due to varying amounts of dark matter admixtures. In contrast to a single-component neutron star of ordinary matter, which yields a unique mass-radius relation, differing amounts of dark matter will fuzzy out such a curve. Thus, first-principle knowledge of this fuzzying, which we will present here, will allow to detect dark matter accumulation.

Moreover, neutron stars with a large amount of dark matter in the interior, also known as mixed neutron stars, will provide potentially spectacular observations. Mixed neutron star binaries are thus a further testbed for dark matter properties, as they are capable of generating exotic signals at gravitational waves experiments. In particular, the so-called tidal deformability, indicating the deformation of colliding neutron stars, has been constrained by LIGO observations of a single binary neutron star merger GW170817 [45, 46]. Along with this LIGO has also measured the "chirp mass" for the same system. These observations have been combined with simultaneous observations from other multi-messenger telescopes, leading to an even more comprehensive analysis [46].

A primary ingredient in understanding such observable effects is the equation of state, which determines the evolution of thermodynamic properties, for both the ordinary component and a potential dark matter component inside the neutron star. Owing to the sign problem [47], lattice QCD, – the methodological mainstay to determine most of QCD physics – is unable to provide a fully controlled equation of state of ordinary matter inside neutron stars. Limited knowledge can be derived from heavy-ion experiments, models, and other methods [48]. However, this has so far been insufficient to provide a complete understanding. In fact, observed neutron star properties are used to constrain the QCD equation of state [14–16, 46], under the assumption that neutron stars are primarily composed of ordinary matter. As a consequence, we will employ here three different candidates for the equation of state of ordinary matter, to represent this uncertainty. The particular choices will be discussed in section 2.

Therefore, while neutron stars may be efficient dark matter accumulators, and consequently dark matter discovery avenues, insufficient knowledge on the accumulated amount coupled with limited control over neutron star ordinary physics present a challenging situation. Progress with this conundrum would therefore be made if at least the dark matter equation of state would be controlled from first principles, reducing the uncertainties on the evolution of one of the two neutron star components. This is possible for WIMPs (and similar) cases, and several studies have been performed in this regard [49, 50].

Owing to the possible existence of large dark matter self-interactions [4–12], QCD-like hidden sectors have recently been in the spotlight as the resulting dark hadrons can

inherently generate the required self-interactions. Such dark sectors thus demand their own equation of state calculations, should neutron stars be used to probe their existence. Similar to ordinary matter, these calculations may, however, also suffer from a sign problem. Fortunately, for some of the cases, like the $Sp(N)$-based QCD-like theories [51–53], the sign problem is absent, and calculations may be possible, or have been done like for a dark matter model based on a two-color QCD-like hidden sector [54], see, e. g., [55–57].

But, both, $Sp(N)$-based and two-color based QCD-like strongly-interacting dark matter scenarios, feature only bosonic hadrons. Such dark matter systems do not experience stabilization by Fermi pressure, and thus gravitational collapse will only be halted when the dark quark substructure becomes relevant. This happens usually only at asymptotically high densities [48], yielding rather a dense dark matter nugget at the center of the neutron star at best, unless strong self-interactions provide stabilization [58].

Therefore any study of mixed neutron stars involving a new QCD-like admixture will benefit from a dark QCD-like sector with fermionic dark hadrons, where the equation of state is accessible in first-principle calculations. Fortunately, such a possibility exists, e. g., a hidden QCD-like sector based on the gauge group $G_2$ [59–61]. It has already been shown that this theory can support pure compact stellar objects [62], which could be interpreted as pure dark neutron stars. Already the one-flavor theory provides a conserved quantum number carried by the fermionic dark baryons [60], and thereby offers a dark matter candidate, alongside a rich spectrum of further states [61]. The two-flavor case furthermore offers dark pions as additional stable states with a Wess-Zumino-Witten-like $3 \to 2$ process, relevant to the dark matter relic density generation mechanism [51–53]. There is also an additional $3 \to 2$ process which converts dark pions into dark fermionic baryons, and allows to build Fermi pressure, from accrued bosonic dark matter, required for neutron star stabilization [62]. $G_2$-QCD therefore provides an ideal testbed for the previously mentioned challenge.

In this work, as a proof-of-principle, we use the available [60–62] equation of state for a dark $G_2$-QCD sector with one flavor[1], and build neutron stars made from both dark matter and ordinary matter. This equation of state has been obtained using lattice simulations [60, 61], and thus fully non-perturbatively. However, it lacks the quantitative precision of ordinary equations of state obtained indirectly, due to limitations in terms of computing time [61]. Nonetheless, as already seen in the studies of pure dark objects [62] the accuracy should be more than sufficient for an exploratory study of the viability of such a scenario.

For ordinary matter, we use a selection of the currently favored equations of state [22, 24, 64] covering a large range of phenomenology. The results will therefore only be quantitatively relevant if either we keep the amount of dark matter limited such that, within errors, neutron star properties remain the same. Or we stipulate that most neutron stars do not contain large amounts of dark matter, but some do, giving a distinct signal differing from ordinary neutron stars when observed. Alternatively, one can interpret our injection of $G_2$-QCD equation of state seriously and demand modification of the ordinary matter equation

---

[1]Since the chiral dynamics is the same for more flavors [59, 61, 63], and up to multiplicities also the spectrum, we do not expect too may non-trivial changes to the equation of state when increasing the number of flavors slightly.

of state to compensate for large modifications in the neutron star properties. However, this is beyond the scope of the present exploratory work, especially in view of the restricted set of available equations of state for the $G_2$-QCD case. We will also assume throughout that the interactions between dark matter and ordinary matter, even at large densities of both, are negligible. This may be justified given that in most strongly-interacting dark matter scenarios the portal couplings to the SM are highly suppressed [65–68].

For the sake of completeness, we discuss briefly our choices of the equations of state in section 2, together with a brief overview of the salient features of $G_2$-QCD. Our approach to building a two-component neutron star is standard [19], and briefly rehearsed in section 3. In section 4 we provide results for a range of dark matter masses and dark matter content. We provide the mass-radius relations, together with maximal and minimal values for both. We also provide the tidal deformability, which is the primary input to the gravitational wave signal for the merging of two neutron stars. Finally, taking our inputs at face value, we interpret our result in terms of limits on dark matter properties or dark matter contents in neutron stars. This part should be considered rather a blueprint than a hard statement, given our limited inputs, but can be improved arbitrarily by improving these inputs. Additional results are available in appendices 6.1 and 6.2.

## 2   Equations of state

The equation of state is the key ingredient for the Tolman-Oppenheimer-Volkoff (TOV) equation [69]. To describe a neutron star consisting of two kinds of fluid we need the equations of state describing the two fluids.

### 2.1   Ordinary matter

We want to cover a large range of possible equations of state for describing the ordinary matter. To this end, we work with the equations of state from [64]. This work uses piecewise polytropes, to interpolate between a low density regime described by nuclear chiral perturbation theory (NChPT) [70] and a high density regime described by perturbative QCD [71]. Equations of state that do not meet the observational constraints from NICER data on the mass, radius and the tidal deformability are omitted. For this study, we use 3 limiting equations (EoS I, II and III) of state to get a complete picture [64]. EoS I is the softest of the three barely reaching the two solar mass limits. EoS II yields a maximal maximum mass. EoS III is the stiffest and while it might be too stiff to describe an ordinary neutron star, we still include it in our analysis as the addition of dark matter to a neutron star usually results in smaller masses and radii. Therefore it might be able to meet observational constraints when considering mixed neutron stars. For the same reason we will employ central pressures larger than the ones stated in [64] that result in the maximal mass. We will see that the addition of dark matter allows for stable solutions at larger central pressures.

## 2.2  $G_2$-QCD

$G_2$-QCD is a QCD-like theory with the gauge group SU(3) replaced by the exceptional Lie group $G_2$ [59, 72]. We consider the situation with one flavor of fermions in the fundamental representation [60, 61], called dark quarks in the following. $G_2$-QCD has, due to its group structure, a number of extraordinary features. Most relevant here is that chiral symmetry breaking occurs already in the one-flavor theory, and that dark baryons can be built with any number of dark quarks [59, 61, 63]. It can also be simulated at all densities using first-principle lattice methods [60] using standard algorithms [47, 73].

A detailed discussion of its spectrum and other properties in the vacuum can be found in [59, 61, 72]. The most relevant features for our purpose here is that the lightest absolutely stable fermionic state is supposedly a three dark-quark state. In addition, there can exist a one dark-quark hadron [59], but it is likely heavier for the dark quark masses employed here, because it is a dark-quark-dark-gluon hybrid state [61, 62]. It is thus supposedly unstable against decay into the three dark-quark state and dark mesons. Note that in $G_2$-QCD dark quarks and dark antiquarks, as e. g. also in Sp(4)-QCD and SU(2)-QCD, cannot be distinguished [63], and thus only Grassmann parity is conserved. In fact, the dark baryonic chemical potential is a Weyl-flavor potential, but serves the same purpose as the baryonic chemical potential in QCD [60, 61]. As a consequence, dark mesons cannot be distinguished from dark glueballs in the one-flavor theory, and thus are only stable if the theory remains in isolation. Hence, we identify the three dark-quark state as the dark matter candidate of our theory. We therefore use its mass also for scale setting.

Details on the equation of state, its determination and properties, can be found in [60–62]. Here, we concentrate on the equation of state at zero temperature, which has been discussed in [61, 62]. Of course, the lattice data is only available at discrete values of the dark baryonic chemical potential. The interpolation between data points is, as for the ordinary matter equation of state, performed using piecewise polytropes to ensure thermodynamic consistency. Following [62], the equation of state is extrapolated to small densities by a free Fermi fluid which follows $n = c(\mu - m_C)^{3/2}$ where $m_C$ is the mass of the dark matter candidate. At high densities we extrapolate the data by a Fermi-Dirac distribution $n(\mu) = n_s/(\exp(a - b\mu) + 1)$ fitted to the last few data points [61, 62]. The upper bound for the equation of state is given by the point where the speed of sound reaches c. Herein $n_s$ is the lattice saturation density, i. e. the situation where Pauli blocking obstructs larger densities. This is a pure lattice artifact but the densities required in the following, as has already been observed previously [62], are small enough for this to be not a relevant issue.

The equation of state has been obtained for two different values of the dark quark mass [61]. The masses are chosen such that the lightest dark hadronic state – the dark pion, which is also unstable due to potential decays to the standard model via messengers – has a mass of 25% and 33% of the dark matter candidate [61].

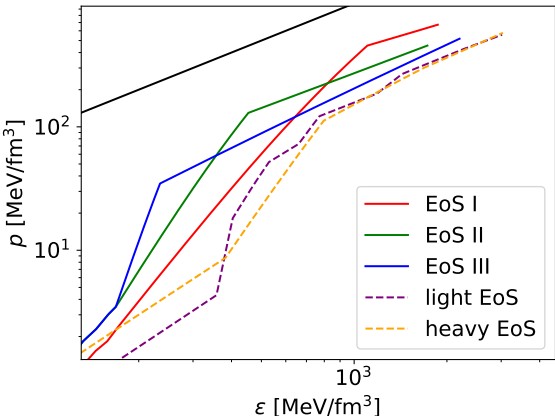

**Figure 1**. The equations of state from [64] and [61] used in this work. The dark matter candidate mass is chosen to be $m_C = 1\,\text{GeV}$ for this plot for comparison. The end of the line indicate the maximum value for the pressure for the respective equation of state used in this work. The black line indicates $\varepsilon = p$.

## 3    Building a two-component neutron star

In this section we will describe the framework used to build a neutron star from the two-fluid TOV-equation [19, 25]. For comparison, we depict the different involved equations of state in figure 3.

### 3.1    Two-fluid TOV

The TOV-equation is a a system of coupled differential equations describing the pressure gradient $dp/dr$ and mass gradient $dm/dr$ derived from the conservation of the energy stress tensor and is consistent with general relativistic effects. Assuming a static, spherically symmetric object in hydrostatic equilibrium, one obtains the ordinary TOV-equation consisting of the pressure and a mass gradient. While usually used for determining ordinary matter properties, a set of two-fluid TOV-equations containing an exotic matter component can be achieved by neglecting the interactions between ordinary and exotic matter components. The following set of equations are the two-fluid TOV-equations written in a dimensionless form [25, 74]. In particular, after switching to natural units, we rescale all dimensionful quantities in units of the mass of the dark matter candidate $m_C$, yielding

$$\begin{aligned}
\frac{dp_O}{dr} &= -(p_O + \varepsilon_O)\frac{d\nu}{dr} \\
\frac{dm_O}{dr} &= 4\pi r^2 \varepsilon_O \\
\frac{dp_D}{dr} &= -(p_D + \varepsilon_D)\frac{d\nu}{dr} \\
\frac{dm_D}{dr} &= 4\pi r^2 \varepsilon_D
\end{aligned} \tag{3.1}$$

where the subscript $O$ refers to the ordinary matter fluid[2] while $D$ stands for dark matter fluid. $\varepsilon$ is the energy density. In the following, lowercase $m$ and $r$ will refer to the mass and radius during the integration while capital letters $M$ and $R$ refer to neutron star properties. Finally, the metric function $d\nu/dr$ is defined for both fluids as

$$\frac{d\nu}{dr} = \frac{(m_O + m_D) + 4\pi r^3 (p_O + p_D)}{r\left(r - 2(m_O + m_D)\right)}. \tag{3.2}$$

The coupled two-fluid TOV equations can be solved using the equations of state of the two fluids. Iterating over the potentially nonidentical central pressures, one obtains the neutron star properties for each combination of central pressures ($p_{0,O}$ and $p_{0,D}$) by integrating the equations until the pressure vanishes. The radius at which the ordinary/dark matter pressure drops to 0 is labeled $RO/RD$. $M_O/M_D$ are the integrated masses of the two fluids respectively and $M_{tot} = M_O + M_D$. The solutions can result in a neutron star with a dark core ($r_O > r_D$) or a dark halo ($r_O < r_D$), which potentially also affects the gravitational wave signal of an inspiral of two neutron stars [32].

Dimensional analysis show that the mass and the radius scale like $1/m_N^2$. A hypothetical neutron star consisting of a 500 MeV neutron would therefore have 4 times the mass and the radius of a realistic neutron star. To some extent, this scaling argument is still present in the two fluid case. A lighter dark matter candidate mass will result in a stronger impact of dark matter on the star. In the limiting cases where the central pressure of one of either the dark or ordinary matter is negligible ($p_{0,D} \gg p_{0,O}$ or $p_{0,D} \ll p_{0,O}$) this scaling argument is perfectly recovered as very heavy dark matter candidates will have no impact on the observables while a light dark matter candidate will result in a compact object dominated by dark matter that can eventually not be interpreted as a neutron star anymore.

## 3.2 Stability

Not every solution of the TOV equations yield a stable neutron star. Small perturbations in the metric field should settle back to the original solution. By solving a Sturm-Liouville problem one can identify the stable solutions by demanding the following criterion [22, 42, 75]

$$\begin{pmatrix} \delta N_O \\ \delta N_D \end{pmatrix} = \begin{pmatrix} \partial N_O/\partial\varepsilon_{c,O} & \partial N_O/\partial\varepsilon_{c,D} \\ \partial N_D/\partial\varepsilon_{c,O} & \partial N_D/\partial\varepsilon_{c,D} \end{pmatrix} \begin{pmatrix} \delta\varepsilon_{c,O} \\ \delta\varepsilon_{c,D} \end{pmatrix} = 0, \tag{3.3}$$

where $N$ is the number of particles of the two types of fluids in the star. This problem can be translated to demanding that both Eigenvalues of the equation in (3.3) are positive. The number of particles can be calculated simultaneously to the TOV equations via

$$\frac{dN}{dr} = 4\pi \left(1 - \frac{2m}{r}\right)^{-1/2} n r^2 dr. \tag{3.4}$$

For polytropes, the number density $n$ is given by

---

[2]Sometimes this is also called baryonic matter. We decide to use ordinary ($O$) as our dark matter is also baryonic.

$$n = \left(\frac{p}{K}\right)^{1/\Gamma}, \tag{3.5}$$

where, K is a constant and $\Gamma$ depends on the polytropic index n like $\Gamma = \frac{n+1}{n}$ [76].

## 3.3 Tidal Deformability

When two compact objects orbit each other, they get deformed by the gravitational tidal field of one another. The tidal deformability $\Lambda$ is a quantity that describes the proportionality between the tidal field and the induced quadrupole deformation. We will work with the unit-less version of the tidal deformability that is connected to the dimensionful tidal deformability $\lambda$ and the dimensionless second Love number $k_2$ via

$$\Lambda = \frac{\lambda}{M_{tot}^5} = \frac{2}{3}\frac{k_2}{C^5}. \tag{3.6}$$

Here, we have introduced the compactness $C = M_{tot}/R$ calculated with the total mass $M_{tot} = M_O + M_D$ and the maximal radius $R = \max(R_O, R_D)$.

Calculating the tidal deformability (eqn. 3.6) requires the second Love number, which can be calculated from the neutron star compactness as

$$\begin{aligned}
k_2 = &\frac{8C^5}{5}(1-2C)^2\left[2+2C(y-1)-y\right] \times \\
&\left\{2C\left[6-3y+3C(5y-8)\right] + \right. \\
&\left. 4C^3\left[13-11y+C(3y-2)+2C^2(1+y)\right] + \right. \\
&\left. 3(1-2C)^2\left[2-y+2C(y-1)\right]\ln(1-2C)\right\}^{-1},
\end{aligned} \tag{3.7}$$

The auxiliary parameter $y$ can be integrated simultaneously to the TOV equation by

$$r\frac{dy(r)}{dr} + y(r)^2 + y(r)F(r) + r^2 Q(r) = 0 \tag{3.8}$$

where

$$F(r) = \frac{r - 4\pi r^3((\varepsilon_O + \epsilon_D) - (p_B + p_D))}{r - 2m(r)} \tag{3.9}$$

and

$$\begin{aligned}
Q(r) = &\frac{4\pi r\left(5(\varepsilon_O + \varepsilon_D) + 9(p_O + p_D) + \frac{\varepsilon_O + p_O}{c_{s,O}^2} + \frac{\varepsilon_D + p_D}{c_{s,D}^2} - \frac{6}{4\pi r^2}\right)}{r - 2m(r)} \\
&- 4\left[\frac{m(r) + 4\pi r^3(p_O + p_D)}{r^2(1 - \frac{2m(r)}{r})}\right]^2
\end{aligned} \tag{3.10}$$

with $m = m_O + m_D$ and $c_{s,O/D}^2 = dP_{O/D}/d\varepsilon_{O/D}$ is the speed of sound [42].

In the late stages of an inspiral event the tidal deformability induces a gravitational wave phase given by a combination of the tidal deformabilities of the two participating

neutron stars. To leading order in the tidal deformability this phase can be determined by the parameter $\tilde{\Lambda}$ as [77]

$$\tilde{\Lambda} = \frac{16}{13} \frac{(M_1 + 12M_2)M_1^4\Lambda_1 + (M_2 + 12M_1)M_2^4\Lambda_2}{(M_1 + M_2)^5} \tag{3.11}$$

where $\Lambda_1$ and $\Lambda_2$ are the tidal deformabilities and $M_1$ and $M_2$ are the total masses of the two neutron stars. As this cannot separate both $\Lambda_i$, we choose conservatively the extreme cases that either one entirely dominates for the limits in $\Lambda$ below.

The applicability of gravitational wave analysis beyond black-hole signals to more complex and exotic systems such as the mixed stars studied here is not thoroughly understood [78–81]. As the nature of this work is exploratory, we will discuss the results of the tidal deformability for all our results for illustrative purposes only.

## 4 Results

In this section we show our results of the TOV-equation. We investigate the three mentioned equations of state for ordinary matter as well as two equations of state for dark matter described by one-flavor $G_2$-QCD at two different values of the bare quark mass labelled as *light* and *heavy*. They correspond to the lightest particle, the dark pion, having a mass of 25% and 33% of our dark matter candidate, respectively [61]. On top of that we investigate the mass $m_C$ of the dark matter candidate from 250 MeV to 4 GeV, and showing three benchmark values for $m_C$ in the figures. This range is well motivated from other studies on self-interacting dark matter and covers all interesting dynamics in the system [52, 53, 82]. Also, fixing the scale is necessary as the combination of Newton's constant and the mass of the neutron in ordinary matter sets an absolute scale. As will be seen, for dark matter masses much smaller or larger than the neutron the solutions can be obtained by scaling arguments. The mass and radius of a single-fluid compact object scale like $m^{-2}$ where $m$ is the scale with which the equation of state is rescaled. This scaling argument is recovered for very light dark matter, as it starts to dominate the neutron star. For heavy dark matter, we obtain an ordinary neutron star dominated by standard model matter. This is not too surprising as, despite all differences in details, the overall scales of $G_2$-QCD are still similar to ordinary QCD [61]. Thus, only if the ordinary matter scale and the dark matter scale are similar a balancing without either side dominating can be expected.

For this intermediate mass range, we solved the two-fluid TOV equation by iterating over both the central pressures of ordinary matter and dark matter and perform the stability analysis. In figure 2 we show the range of central pressures that we investigated. Gray points indicate a stable solution to the TOV-equation. One immediate observation is that the addition of dark matter makes it possible to sustain larger central pressures for ordinary matter. This is particularly interesting as this grants access to new regimes of equations of state that could potentially have interesting phenomenology. Moreover, heavier dark matter results in larger possible central pressures which can also be inferred from scaling arguments (see section 3.1).

As an additional constraint, we want to investigate under which circumstances the objects considered in this paper are compatible with astronomical observation. Because

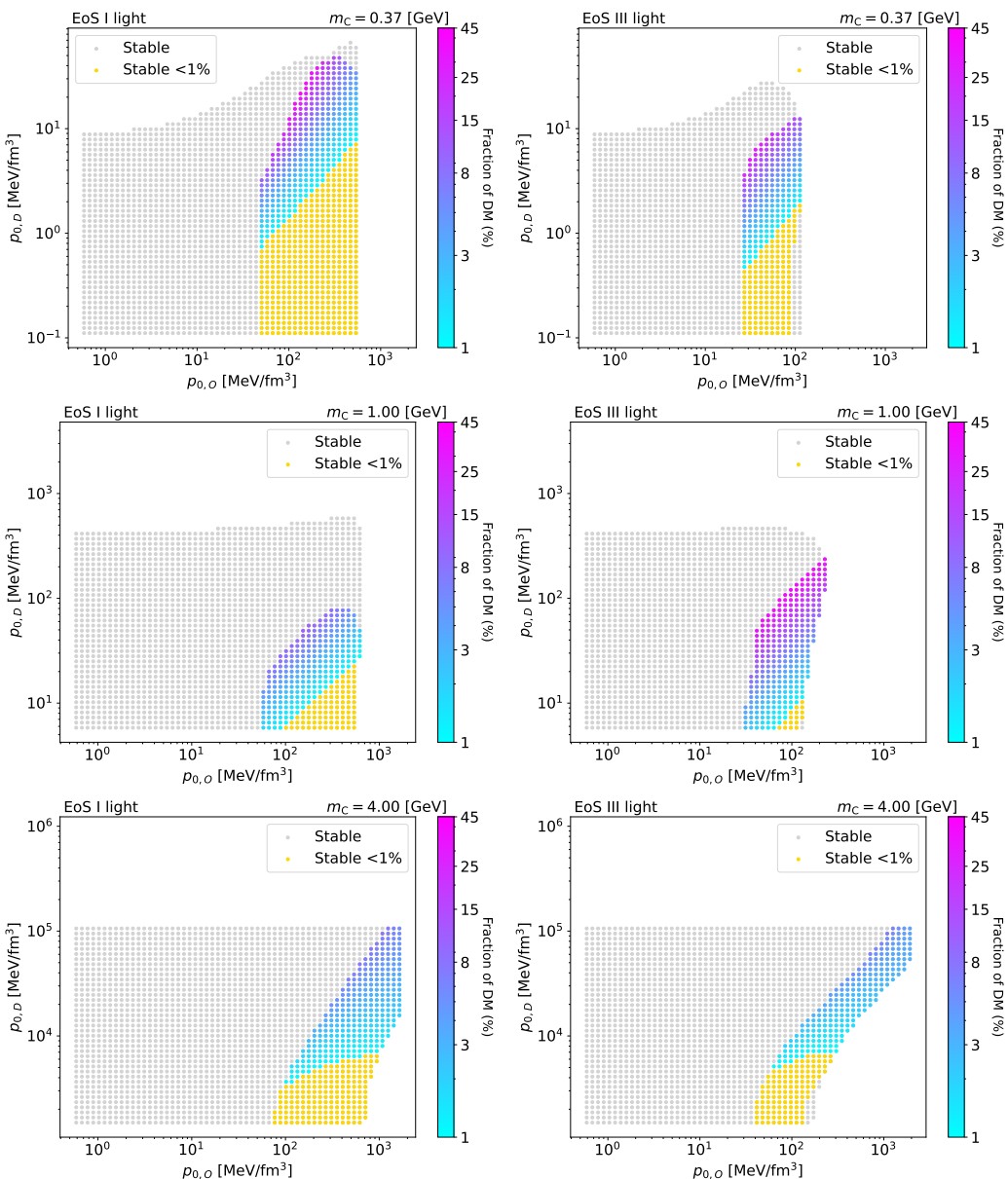

**Figure 2**. The investigated set of central pressures of ordinary matter ($p_{0,O}$) and dark matter ($p_{0,D}$). Gray points indicate stable solutions outside the constraints chosen in (4.1). Colored points indicate stable solutions that meet the constraints at >1% dark matter fraction for the color coding and <1% in yellow. The dark matter mass increases from 0.37 GeV (top) and 1 GeV (middle) to 4 GeV (bottom). The left-hand side shows the results for ordinary matter equation of state I, and the right hand side for III. Both are for the light dark matter equation of state.

this is an exploratory study, we intentionally choose loose bounds on the mass, the radius

and the tidal deformability given by

$$0.7 M_\odot < M_{\text{tot}} < 2.2 M_\odot$$
$$10 \text{ km} < R_O < 16 \text{ km} \tag{4.1}$$
$$\Lambda < 2000$$

where $M_{\text{tot}} = M_O + M_D$. Here, the bound on the radius only applies for the radius given by the ordinary matter as observation constraining the radius of neutron stars rely on electromagnetic radiation [16], which is expected to pass a potential dark matter halo unhindered. At the same time, mass is determined from gravitational effects, and will thus be provided by both components equally. In both cases the limits are chosen such as to loosely represent a conservative bound on astrophysically obtained result [16]. Finally, the Love number is bounded from gravitational waves measurements [45]. We chose a more conservative bound to account for effects of dark matter and because the attribution of it to individual stars is difficult [46]. The limit is therefore representing the most extreme possible cases. Again, this quantity is determined by a combination of dark matter and ordinary matter. The stable solutions to the TOV-equations that meet the criteria above are color coded by the amount of dark matter they contain $M_D/M_{\text{tot}}$. The maximal value of the color coding (45%) is given by the largest amount of dark matter consistent with the constraint within the three dark matter candidate masses shown. On top of that, we separate the cases where the neutron star contains only up to 1% dark matter mass and indicate these solutions by yellow points[3]. Different publications use different realistic upper bounds for the amount of dark matter in neutron stars [32, 83–85]. With our choice of 1% we want to ensure that the resulting compact objects can actually be identified with neutron stars. We will consider results that exceed this bound as non-standard compact stellar objects. As noted in the introduction, if the latter would be the dominating population, our reliance on a fixed QCD equation of state may become dubious.

In figure 3 we show the mass radius relation for all stable neutron stars for 3 different values for the dark matter mass and for two of the three ordinary matter equations of state. These choices are representative. We thus relegate here and hereafter results for other setups to appendix 6.1. As expected, the impact of the dark matter is larger for light dark matter where we achieve very heavy objects that contain large amounts of dark matter. Although these are unlikely to be common neutron stars, they could potentially be interpreted as heavy non-standard compact stellar objects. This is an interesting possibility, as stiff equations of state are usually hard to obtain. By increasing the dark matter mass, its impact becomes more subtle in the sense that the resulting objects could very well be neutron stars with small dark matter parts.

We show contributions from the individual parts to the mass in figure 4 and for the radius in figure 5. We again also illustrate if the dark matter contribution to the total mass is limited to 1% of the total mass. The first important observation from figures 3-5 is that changing the dark matter mass by an order of magnitude has much more severe impact than

---

[3]The color scheme of the points in this plot will be used throughout the paper.

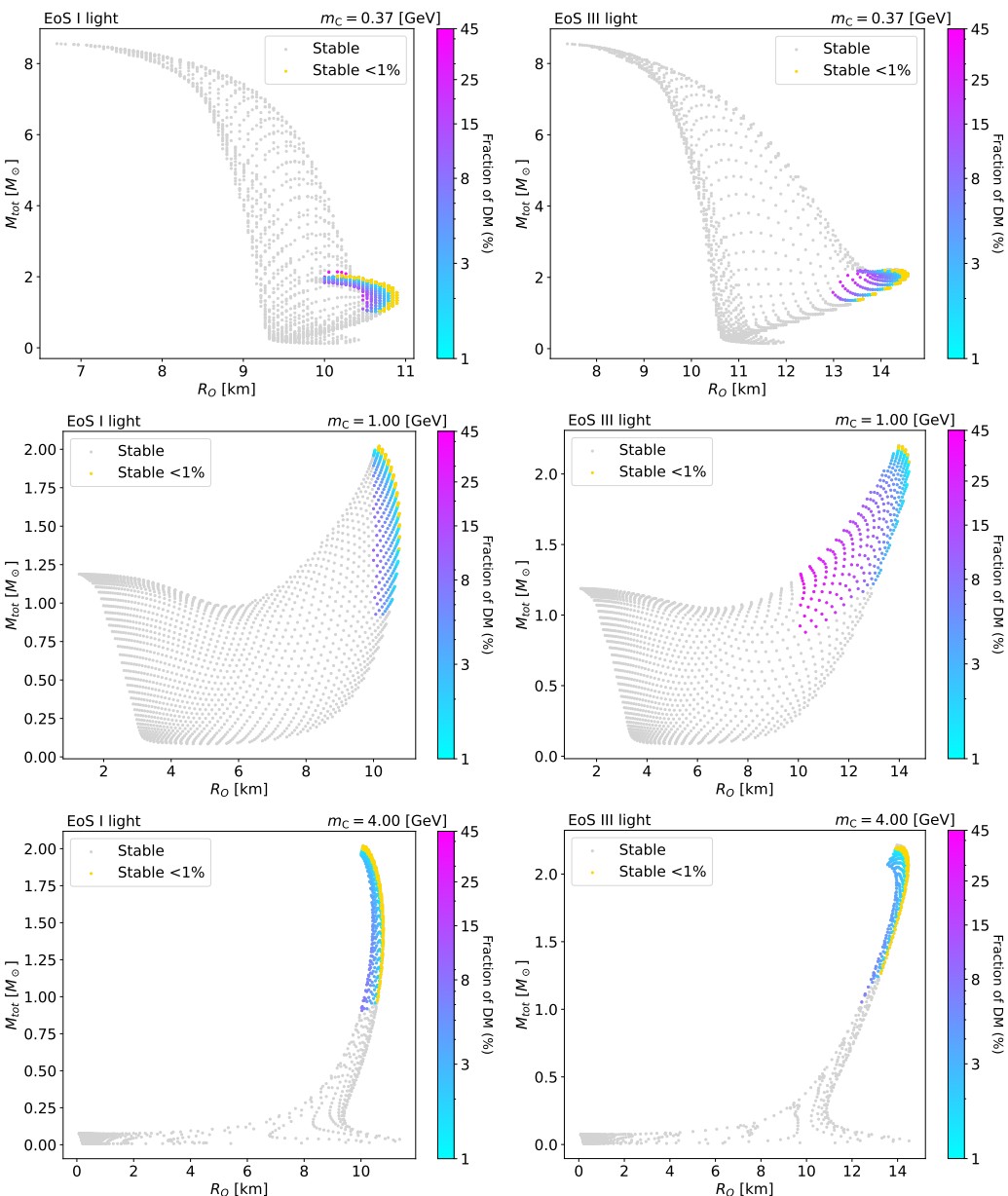

**Figure 3**. The mass-radius relation between the observable ordinary radius $R_O$ in km and the total observable mass $M_O + M_D$ in solar masses in the same color-coding as figure 2. The left-hand side shows the results for ordinary matter equation of state I, and the right hand side for III. Both are for the light dark matter equation of state.

changing the equation of state, except for the radius. The observable (maximal) radius is basically entirely dominated by the ordinary matter equation of state.

The contribution of dark matter to the total mass is, on the other hand, primarily dominated by the mass of dark matter particles, with a decreasing contribution the larger the mass. The effect is stronger than linear, and can be traced back to the overall scaling

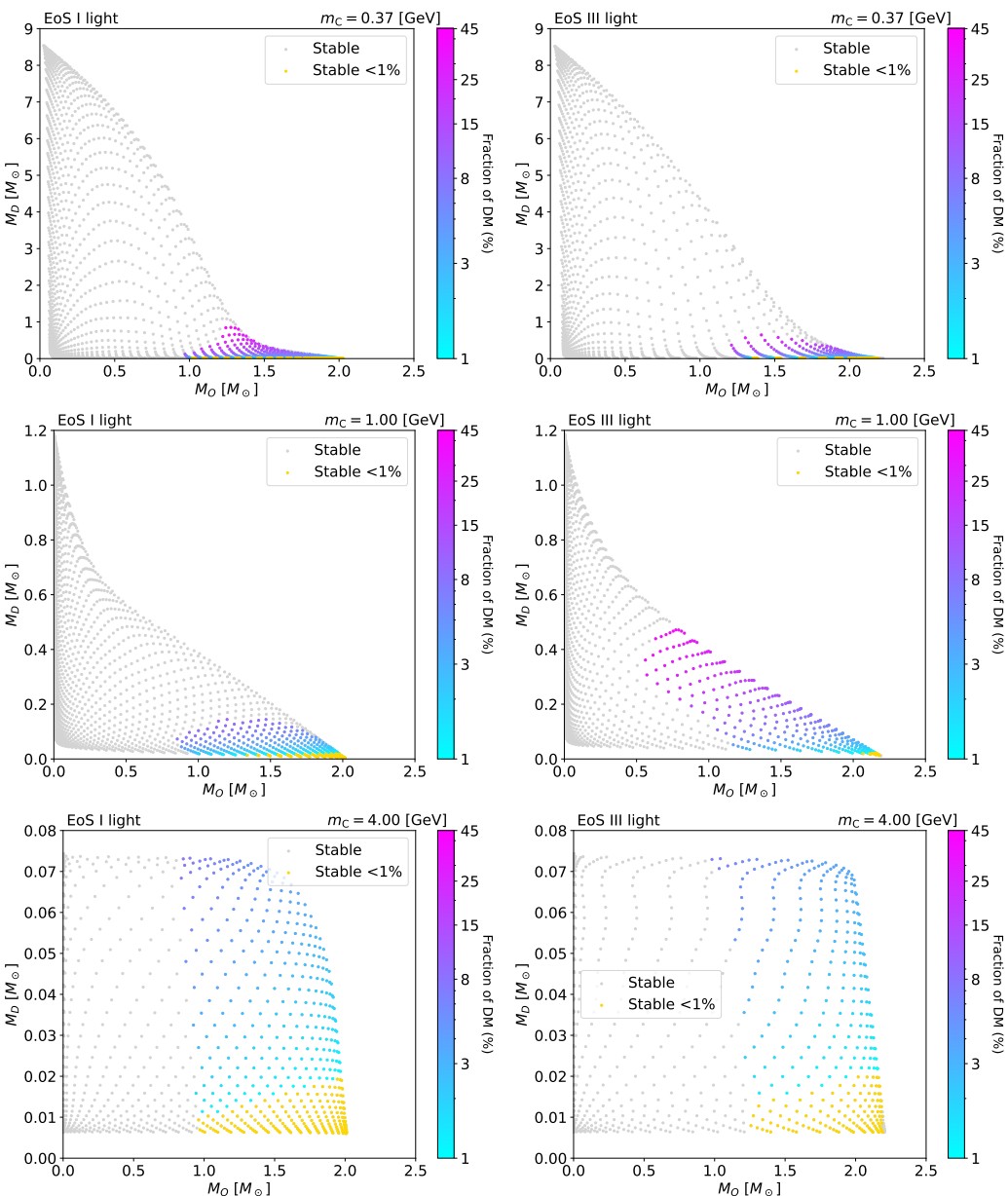

**Figure 4**. The decomposition of the total mass in ordinary mass $M_O$ and dark mass $M_D$ in solar masses in the same color-coding as figure 2. The left-hand side shows the results for ordinary matter equation of state I, and the right hand side for III. Both are for the light dark matter equation of state. Note the different scales for $M_D$.

of the equation of state of the dark matter by the dark matter mass. Still, once the dark matter mass is limited to at most 1% of the total mass, this effect becomes almost negligible. Note that very light dark matter allows for stable neutron stars with more than eight solar masses. While it is unlikely to be actually possible to stabilize the star at such masses, it indicates that the very rare observation of very heavy neutron stars could be linked to

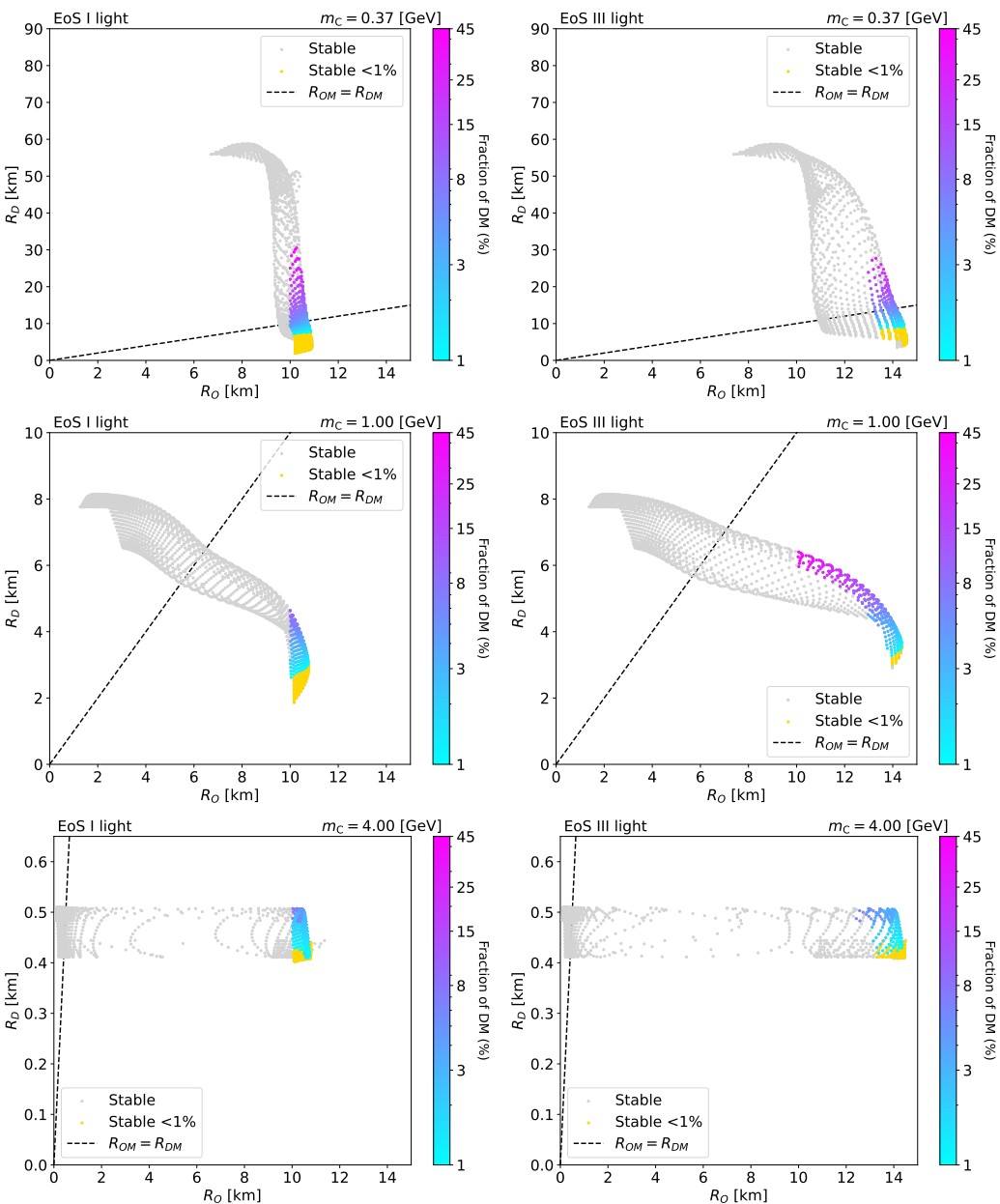

**Figure 5**. The observable radius of the ordinary $R_O$ in comparison to the radius of the dark matter radius $R_D$ in km in the same color-coding as figure 2. The left-hand side shows the results for ordinary matter equation of state I, and the right hand side for III. Both are for the light dark matter equation of state. The dotted line indicates where $R_O = R_D$. Solutions above indicate a dark matter halo scenario while solutions below indicate a dark matter core scenario.

a dark matter component. Especially as very stiff equations of state for ordinary matter are not simple to obtain [48]. In particular, our full dark matter equation of a QCD-like dark sector also does not in itself sustain very heavy neutron stars [62]. This may suggest that a separate investigation of the very heavy neutron stars, allowing for modifications of

the ordinary matter equation of state, may be a worthwhile effort. This is an important hint, which could only be obtained from knowing a QCD-like equation of state, as is at our disposal.

The effective radius of the dark matter contribution is again strongly dominated by the dark matter mass, with the radius being inversely and non-linearly dependent on the dark matter mass. Especially for dark matter heavier or as heavy as a neutron, dark matter forms a core, while for very light dark matter a substantially extended dark matter halo is observable. In combination with the aforementioned results on the mass, this gives an interesting possibility. For light dark matter and heavy neutron stars, the dark matter halo in a neutron star merger should interact earlier than the ordinary matter core. That could give a modification of the gravitational wave signal [32]. In combination with relaxing the stiffness of the ordinary matter equation of state, this would be a highly interesting future research project.

## 5   Summary

In this work we studied the impact of a strongly-interacting dark matter component on neutron star properties, using $G_2$-QCD. The equation of state for the dark fluid was obtained at finite density from first principles using lattice field theory [60–62]. We used the two equations of state at our disposal, differing in their mass scales, and found that they have similar influence on the resulting properties of the mixed star. On the standard model side, we covered a large range of possibilities for the ordinary matter equation of state coming from the interpolation between low and high densities using piecewise polytropes [64].

As is well known from investigations of the ordinary matter equations of state, they all result in different mass-radius-relations. The addition of dark matter has similar effects on all three of them. For each combination of equations of state we varied the mass of the dark matter candidate and the central pressure of the ordinary fluid and dark fluid and performed a stability analysis. The addition of dark matter, in general, allows for a larger possible central pressures which should be kept in mind when investigating equations of state. As expected, we find that light dark matter has a stronger effect on the properties of the neutron star and that by tuning the ratio of central pressures one is also able to find solutions that fulfill the experimental bounds. For some combination of the dark matter candidate mass and the ratio of central pressures we find results that do not resemble neutron stars ($M_{DM}/M_{tot} > 1\%$) but could provide an explanation for non-standard compact stellar objects. An astrophysical observation of exotic compact objects also in gravitational wave experiments would allow for the discrimination of the composition of such objects. We leave this for future explorations.

It is noteworthy that, for the first time, this investigation used an equation of state from first principles for describing strongly-interacting dark matter in a two component neutron star. While the impact on, e. g., the mass-radius-relation may be similar to using an effective model, first principle calculations of a UV-complete confining theory that is also capable of addressing small-scale structure problems are an important step towards unveiling the true nature of dark matter.

## Acknowledgements

We are grateful to J. Schaffner-Bielich and V. Sagun for helpful discussions. Y. D. has been supported by the Austrian Science Fund research teams grant STRONG-DM (FG1). S. K. is supported by the Austrian Science Fund research teams grant STRONG-DM (FG1) and project number P 36947-N. S. K. thanks the University of Würzburg for hospitality through the RTG 2994 program, while part of this work was completed.

## 6 Appendix

### 6.1 Results for the heavy dark matter equation of state

In this appendix, we compile a set of results for the heavy equation of state for the dark matter sector, as well as for the equation of state II. We did not use them in the main text as they do not substantially change the overall picture. This can be seen for the mass-radius relation shown in figures 6 and 7, which need to be compared with figure 3.

### 6.2 Results for the Tidal Deformability

For completeness, we show in figure 8 the results for the tidal deformability $\Lambda$, which is directly calculated from $k_2$ using eqn. 3.6 using the formalism described in section 3.3. We choose these equations of state to showcase the general behavior. We see similar features as in [24]. The data points compatible with our constraints with a dark matter mass fraction $<1\%$ are essentially equal to the results for pure neutron star calculations. Allowing for a larger dark matter mass fraction, you obtain results with larger values of $k_2$ at smaller compactnesses. Given the scaling of $\Lambda$ with the compactness as $C^{-5}$, this will result in much larger values for $\Lambda$, which can also be seen on the right-hand side of the plot. We have already discussed the applicability of these results for gravitational wave signals in section 3.3. In the case shown here, the results close to a pure dark star yield slightly larger values for $k_2$ at larger values for C which in general results in smaller $\Lambda$. These kind of objects will become interesting if the merger of exotic compact objects is detected in a gravitational wave experiment.

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
