# Peer review of "Strongly Interacting Dark Matter admixed Neutron Stars"

_SciPost Physics Core_

## Round 2 · Referee Report · Anonymous (Referee 1) · 2025-9-19

The referee discloses that the following generative AI tools have been used in the preparation of this report:
I used the generative AI tool only to polish the language of my review. The scientific evaluation, analysis, and conclusions are fully my own.
Strengths
2-Introducing a lattice-derived EOS for dark matter into two-fluid neutron star modelling is a novelty and useful blueprint for future studies.
3-The main qualitative conclusions, light $m_c$ tends to halos while heavy $m_c$ tends to cores; sub-percent DM fractions yield tidal properties close to ordinary neutron stars, are plausible and well illustrated.
4-The authors perform an extended investigation by varying the central pressures of both ordinary and dark matter components and systematically scanning the parameter space according to their chosen criterion [Eq. (4.1)].
5-By considering a broad range of observables, the study provides a useful framework that future works can employ for comparison and further refinement.
6-The study employs three different ordinary matter EOSs with varying stiffness, allowing for an exploration of model dependence in the ordinary matter sector.
7-For the dark matter sector, two distinct EOSs are considered, which broadens the scope of the analysis and strengthens the robustness of the results.
8-The authors carry out a careful and detailed analysis of the ordinary and dark components separately, specifically the masses (M_O,M_D) and radii (R_O,R_D), which is particularly valuable in the context of two-fluid dark-matter–admixed neutron stars.
Weaknesses
1-Observational constraints (mass, radius, tidal deformability) not applied tightly enough or not shown under stricter cuts. 2-Tidal deformability analysis is too limited (only the softest EOS and the DM mass corresponding to the core configuration have been presented). 3-Figures are not easily readable, and the corresponding texts are not clarifying (the connection to M–R phenomenology is unclear.) 4-While the broad range of observables is valuable, it does not establish the consistency of the selected parameters with current astrophysical measurements. As a result, the dark matter parameters are not effectively constrained by the presented calculations. Consequently, the main conclusion stated in the abstract, that dark matter masses of a few hundred MeV to a few GeV are consistent with the latest neutron star observations, appears overstated, since the latest observational constraints were not actually incorporated into the analysis. 5-Lack of standard fixed central-pressure ratio or fixed dark matter fraction sequences for a full M-R line, which are widely used in neutron star literature. 6-Unclear treatment of dark matter self-interaction (implicit in lattice EOS but not clarified). 7- Lack of any report about the speed of sound. 8-In one instance, the authors cite around 30 references immediately after the sentence “The amount of dark matter may vary depending on the age, history, and location of the neutron star, but most estimates conclude that it is very unlikely to exceed 1% of the total neutron-star mass.” This is unusual and should be reconsidered.
Report
With this clarification, the manuscript will offer a clear and robust reference for dark matter admixed neutron stars using first-principles dark-sector input. The additions do not change the scope but improve clarity and completeness.
Requested changes
1- First of all, I would like to emphasize that the authors should compare their results directly with the most recent observational constraints on neutron stars if they intend to propose their parameter space for dark matter (even if it is only the dark matter particle mass) as a consistent range. In particular, they need to present both mass–radius and tidal deformability relations in a way that clearly illustrates the behavior of their models relative to the data. They may clarify this with some plots with exact observational constraints included, or by providing tables or texts which explain explicitly which range of parameters and which EOSs are consistent with the observational data (For example, $\Lambda_{1.4}$<580). This is the main missing part of the paper. 2-The authors should elaborate on the role of dark matter self-interactions in their framework. 3-The authors should at least provide some comments on the speed of sound in the employed EoSs. 4-The authors state that they interpolate piecewise polytropes between NChPT at low densities and pQCD at high densities. However, the maximum energy density reached in their models is well below the regime where pQCD is valid. This statement should therefore be revised to accurately reflect the interpolation procedure actually used. 5-On page 7, the authors claim that the crust would hardly affect the radius. This is not correct, as the crust has a non-negligible impact on neutron star radii. The statement about not including a crust should be modified accordingly. 6-For clarity and consistency, I recommend revising the use of terminology and abbreviations. Some common terms (e.g., "equation of state," "neutron stars") are repeated many times without abbreviation, while certain abbreviations (e.g., WIMPs) appear without definition, or are defined only after their first occurrence (e.g., NChPT). Defining each abbreviation upon its first use and employing it consistently thereafter would improve readability and flow. 7-Correct the thermodynamic identity to $n\mu = \varepsilon + p$ (instead of $n\mu = p\varepsilon$) before Eq. (6.1). 8-Correct the expression for the chemical potential to $\mu = c + \tfrac{\Gamma}{\Gamma - 1} K n^{\Gamma - 1}$, not $\Gamma/(\Gamma+1)$. 9-Review the manuscript for typographical errors such as “we use choose …” or “is a a system …” and correct them throughout.
Recommendation
Ask for minor revision

---

## Editorial Decision

unknown